# Risk of COVID-19 and Cost Burden in End-Stage Renal Disease Patients and Policy Implications for Managing Nephrology Services during the COVID-19 Pandemic

**DOI:** 10.3390/healthcare10122351

**Published:** 2022-11-23

**Authors:** Seda Behlul, Macide Artac Ozdal

**Affiliations:** Department of Health Management, Faculty of Health Sciences, European University of Lefke, TRNC-10 Mersin, Lefke 99770, Northern Cyprus, Turkey

**Keywords:** COVID-19, pandemic, haemodialysis risk, cost burden, management

## Abstract

The aim of this study was to evaluate the risk of COVID-19 in end-stage renal disease (ESRD) patients, the cost burden of the COVID-19 pandemic on the management of ESRD and the cost of catheter infections. In this multicentre, retrospective study, data were obtained from the records of four dialysis centres providing care for ESRD patients in Northern Cyprus. Of the 358 ESRD patients that were receiving haemodialysis (HD) 13 were diagnosed with COVID-19. The average cost of HD treatment per patient was $4822.65 in 2019 and $3759.45 in 2020 (p ≤ 0.001). The average control cost of HD treatment per patient was $618.80 in 2019 and $474.03 in 2020 (p ≤ 0.001). The outpatient treatment costs of catheter infections were not significantly different in 2019 (before) compared to 2020 (after) the pandemic ($54.61 in 2019 compared to $54.74 in 2020, p = 0.793). However, the inpatient treatment costs were significantly greater before the pandemic compared to after the pandemic ($315.33 in 2019 compared to $121.03 in 2020, p = 0.015). The costs for monitoring COVID-19 transmission in patients having ESRD management were significantly higher in HD compared to in peritoneal dialysis (PD) and transplants. Since there is a high risk of transmission of infections in the hospital environment during a pandemic, it is important to implement alternative ESRD management methods, such as enhancing transplants in populations, switching to PD, and implementing home dialysis programmes to reduce the risk of infection and associated complications, as well as the health costs associated with infection monitoring.

## 1. Introduction

The coronavirus disease, 2019 (COVID-19) pandemic has been affecting populations since the end of 2019, after the emergence of the severe acute respiratory syndrome coronavirus 2 (SARS-CoV-2) infection in Wuhan, China [1]. Various undesirable consequences have been caused by the pandemic since healthcare resources have had to be reallocated for the management of COVID-19 cases in many parts of the world. This shift in resources has particularly affected the continuation of treatment and management of chronic diseases in developing countries. Evidence shows that those at greater risk are the elderly and those with chronic diseases, such as diabetes, chronic kidney disease, chronic heart disease and others. It is, therefore, crucial to target the care of chronic patients in conditions that do not increase the risk of COVID-19 transmission [2,3]. After it was first detected in December, 2019, the COVID-19 virus spread quickly across the world, becoming a pandemic in March, 2020. Various public health policies were implemented towards the control of COVID-19 infections around the world. COVID-19 caused an increase in mortality and fatality rates globally, with mortality rates ranging from 0.2 to 8% and care fatality rates between 2 and 3%. It is crucial to detect the factors that determine the severity of infection to stratify risk. This is important in terms of efficiently allocating resources to healthcare organizations and formulating effective public health policies [4].

Chronic kidney disease (CKD) is a disease that is diagnosed by the existence of reduced functioning and structural damage in the kidneys. The effective and efficient treatment of CKD is crucial in terms of preventing the disease from progressing to the terminal stage, known as end-stage renal disease (ESRD) [5]. Once ESRD develops, the quality of life is inversely affected and risk of mortality increases. The treatment and management of the condition become difficult with the requirement of advanced methods, such as haemodialysis (HD), peritoneal dialysis (PD), and kidney transplantation (Tx) [6]. As well as the significant impact of CKD on morbidity and mortality levels in populations, it exerts a large burden on countries’ economies and health systems. A range of challenges have been experienced in developing and developed countries in terms of providing high-quality, effective and affordable kidney replacement therapy for patients with ESRD [2]. CKD can progress into ESRD, which leads to a significant burden on public health. This stresses the importance of diagnosing CKD at early stages to prevent its progression into ESRD, which requires costly treatment with renal replacement therapy (RRT). Around the world, most CKD patients are living in low- and middle- income countries, which have scarce resources that can be allocated to healthcare, particularly for the management of chronic diseases. Although the majority of countries have improved access to RRT services, many patients are still experiencing problems in affording RRT [7]. There has been a rapid increase in the rates of ESRD, which is increasingly exerting a high burden on health and healthcare systems. This reduces the affordability of providing essential prevention services for those at risk and treatment services for those who have already developed ESRD, particularly in low- and middle-income countries [8]. Dialysis treatment has a high-cost burden. The cost of dialysis includes several components, including disposables, dialysis machines, accommodation, water, health care staff costs, and electricity. There are also additional costs for medications, transportation to the hospital, hospital admission due to complications, and any required intervention [9].

When patients receiving HD are infected with COVID-19, they require intensive care, because they are immunocompromised, tend to have many comorbidities and also mostly have close contact with other patients when having HD [10]. Despite the decline in the mortality rate in ESRD patients in the United States since 2001, there has been an increased risk of morbidity and mortality in ESRD patients due to COVID-19 infection. due to their weak immune systems and possible multiple comorbidities [11]. Evidence shows that patients having hemodialysis treatment have a greater risk of developing infections as a complication of the treatment they are receiving. The first death due to COVID-19 that occurred in the US was observed in a hemodialysis patient. Therefore, this prompted studies that tried to define the impacts of COVID-19 on people undergoing HD [12,13,14]. In a study on the mortality rates in patients receiving HD in New York City, the mortality rate was found to be 31%. In patients who required mechanical ventilation, the mortality rate was 75%. Studies in Europe have also revealed similar mortality rates to those in New York City [13,15]. HD is the most common treatment that ESRD patients receive, because not all such patients are eligible for Tx and the kidney supply for ESRD patients who are eligible for transplantation is not adequate [16].

This study focused on assessing the risk of COVID-19 in ESRD patients receiving RRT, the cost burden of COVID-19 in ESRD patients, and the impact of the pandemic on the cost burden of ESRD in Northern Cyprus. The objectives of the study included: (i) evaluation of the risk of COVID-19 in ESRD patients; (ii) assessment of the cost burden of the COVID-19 pandemic on the management of ESRD patients; (iii) assessment of the difference between the direct medical costs of HD treatment before and after the COVID-19 pandemic; (iv) assessment of the cost of catheter infections, which is one of the most important risks of HD; and (v) providing suggestions for improving the management of ESRD.

## 2. Materials and Methods

This study focused on ESRD patients receiving RRT in Northern Cyprus. The study assessed the risk of COVID-19, the cost burden of the COVID-19 pandemic, and the risk and cost burden of catheter infections in ESRD patients. The assessment of the COVID-19 risk was done by analysing the number of COVID-19 cases in different RRT groups between March, 2020, and October, 2021. The cost burden of COVID-19 was assessed by calculating the costs of diagnostic methods used for identifying COVID-19 (Polymerase chain reaction (PCR) and computer tomography (CT)) between March, 2020, and March, 2021, in patients diagnosed with ESRD and undergoing RRT. The direct costs of HD treatment and the cost of catheter infections, a significant complication of HD, were comparatively calculated in patients having RRT in 2019 and 2020. PCR, CT, HD treatment costs, HD laboratory and imaging costs, and catheter infection costs were calculated in dollars ($) (1 USD = 5.68 TL, inflation rate = 11.84% in 2019; 1 USD= 7.01 TL, inflation rate = 14.16% in 2020).

### 2.1. Study Design and Data Collection

The study was carried out as a retrospective cohort study, which included data collected from the patient records of 4 different dialysis centres providing ESRD care for patients. The data for the study were collected from the clinical, administrative, and financial records of hospitals affiliated to the Health Ministry in Northern Cyprus.

### 2.2. The Research Questions and Significance of the Study

The research questions and motivation factors were as follows:What are the costs of COVID-19 diagnostic methods, such as PCR and CT?What are the costs of COVID-19 diagnostic methods according to RRT?What was the incidence of COVID-19 in ESRD patients between March 2020, and October 2021?What were the comparative HD treatment costs, laboratory costs, and catheter infection costs of HD patients in 2019 and 2020, before and after the outbreak of the COVID-19 pandemic?

### 2.3. Data Analysis

The statistical analysis was performed by using the Statistical Package for the Social Sciences (SPSS v27). A p-value less than 0.05 was accepted as significant. Descriptive analyses were performed to assess the characteristics of ESRD patients. The number of COVID-19 in ESRD patients receiving of RRT and the costs for monitoring COVID-19 infections. Inferential analyses were used to compare costs per patient, based on patient characteristics and RRT. PCR and CT diagnostic expenditures in dollars were analysed by RRT using ANOVA models. The differences between HD treatment costs, HD laboratory costs, outpatient treatment costs and inpatient treatment costs of catheter infections in 2019 and 2020 were evaluated with the Wilcoxon Signed-Ranks Test.

### 2.4. Ethical Approval

The study methodology was approved by the European University of Lefke Ethics Committee, with number ÜEK/57/01/12/2021/02, and the Ethics Committee of Dr. Burhan Nalbantoğlu State Hospital, with the number 08/21.

## 3. Results

Table 1 presents the characteristics and the treatment information of the patients receiving RRT. A total of 310 ESRD patients were registered in 2019 in the nephrology services in the country, a total of 326 patients were registered in 2020. Among all ESRD patients, 252 (81.3%) were receiving HD in 2019 and 257 patients (78.83%) were receiving HD in 2020. A total of 29 (9.4%) and 30 (9.2%) patients were receiving PD in 2019 and 2020, respectively. There were 29 (9.4%) and 38 (11.65%) patients who received a kidney transplant in 2019 and 2020, respectively. The majority of ESRD patients were male (64.2% in 2019 and 65.6% in 2020). In 2019, 70 patients (22.6%) were newly diagnosed with ESRD (diagnosed in the particular year) and 73 patients (22.4%) were newly diagnosed in 2020. A total of 57/310 (18.4%) patients died in 2019 and 49/326 (15.0%) patients died in 2020.

A total of 4 centres provided RRT, whilst the main centre, located in the only General Hospital in the country, provided both regular dialysis care for those residents in the capital, as well as for all patients experiencing complications. About half of all dialysis machines (49.25%) were located in the main centre. Table 2 shows the number of haemodialysis machines in the centres. Centre A had a total of 39 machines, and 23 of these 39 machines were used regularly by patients. After the outbreak of the COVID-19 pandemic, the machines’ capacities were adjusted accordingly, and one machine was allocated for those who were in close contact with those with COVID-19 (those in isolation), and one machine was allocated for those with COVID-19. These machines were also located in the main centre in Centre A. Those diagnosed with HD in other centres were required to travel to the main centre to have HD in the case of COVID-19 infection or when they were in isolation.

The COVID-19 cases in ESRD was studied for the period between March 2020, and December 2021 (Table 3). In total, 358 ESRD patients were receiving treatment in nephrology services during the study period, with 285 HD patients being followed up in 4 centres. ESRD patients’ diagnosis with SARS CoV-2 was identified based on the positive cases registry of the Ministry of Health. COVID-19 was detected in 13 of the 358 patients receiving RRT. COVID-19 cases were only detected in patients having HD (*n* = 285). Out of 13 patients with COVID-19, 9 were hospitalized, and 4 received outpatient treatment at home or in hotel quarantine. The treatment of 5 out of 9 patients continued in the Intensive Care Unit (ICU) and the haemodialysis treatment of these patients continued in the ICU unit. However, these 5 patients died (38.5% of HD patients who had COVID-19) in the ICU. The other 4 patients infected with COVID-19 who were hospitalized in the pandemic hospital continued to have HD sessions in isolation. Patients staying in a hotel or home quarantine were transported to the hospital with vehicles specified by the state or by their vehicles and continued HD sessions in an isolated environment determined according to pandemic rules. The mortality rate due to non-COVID reasons was 27.71% in HD patients (79 deaths in 285 HD patients) and 31.03% in PD patients (9 deaths in 29 PD patients).

The analysis of COVID-19 vaccination status in the HD patients showed that 51.9% of patients (148 patients) had had 4 doses of vaccine, whilst 38.2% of them (109 patients) had 3 doses and 7.02% of patients (20 patients) had 2 doses of vaccine by the end of the study period. Only one patient had one dose of vaccine and 7 of them were not vaccinated. All Tx (44 patients) and 28 out of 29 PD patients received 4 doses of COVID-19 vaccine.

Table 4 presents the costs of COVID-19 management using PCR and CT diagnostic methods for ESRD patients based on the type of RRT they were receiving between March, 2020, and March, 2021. Among 326 ESRD patients observed in the study period, the mean cost of PCR was $63.88 (57.10–70.65) per patient and the mean CT cost was $2.88 (2.22–3.53) per patient. The mean PCR cost for the patient groups treated with different RRT was highest in patients receiving HD, compared to those receiving other RRT [HD: $75.53 (67.72–83.34); PD: $36.75 (23.8–50.12); Tx: $8.67 (3.82–13.52)]. When the mean CT costs in these patient groups were compared, HD patients were found to have the highest CT costs, with $3.51 (2.71–4.31) per patient. When the PCR and CT cost of these patients were analysed, the COVID-19 diagnosis, including the PCR and CT costs, were determined to be highest in the HD group. A statistically significant difference was found between the PCR and CT mean costs in patients receiving HD, PD, and TX (*p* < 0.05).

The direct medical costs of the studied patients in 2019 and in 2020 are presented in Table 5. The HD treatment costs consisted of each bicarbonate fee calculated according to the Health Institutions Fee Schedule Regulation. The laboratory costs of the HD patients consisted of general laboratory costs, routine imaging, and laboratory tests. The total cost of catheter infections included the laboratory tests performed for catheter infections, imaging, medications given for treatment, and the patient’s outpatient or inpatient treatment (hospital-bed care fees) costs. A total of 146 patients were followed up in the main centre in 2019 and 169 patients in 2020. Among the patients who continued HD in the main centre, catheter infection was detected in 34 patients in 2019 and 63 patients in 2020. In 2019, for 252 HD patients, the mean cost of HD was $4822.65 (4557.89–5987.43) and the mean general laboratory cost was $618.80 (592.2–645.3). For the same time= period, catheter infection was detected in 34 patients out of 169 HD patients followed up in the main centre. For these patients, the mean cost of outpatient treatment was $54.61 (34.29–74.94) and the mean cost of inpatient treatment was $315.33 (131.34–499.32). In 2020, for 257 HD patients, the mean cost of HD was $3759.45 (3573–3945.8) and the mean general laboratory cost was $474.03 (455.53–492.54). Catheter infection was detected in 63 of 169 HD patients followed up in the main centre. The cost of outpatient treatment for patients with catheter infection was determined as $54.74 (40.72–68.76) and the cost of inpatient treatment was determined as $121.03 (66.28–175.78). The mean cost of total HD treatment, HD control costs, the total cost of catheter infection, outpatient treatment of catheter infection, and inpatient treatment cost of catheter infection were significantly lower in 2020 compared to 2019 (*p* ≤ 0.05).

The costs of care given to ESRD patients were separately studied for the periods of 2019 and 2020 separately in terms of HD treatment costs, HD control costs, outpatient treatment, and inpatient treatment costs of catheter infections (Table 6).

The cost of care provided to ESRD patients was also studied, based on the time of disease diagnosis. In 2019, the costs of HD treatment, the costs of HD treatment control, and inpatient treatment costs of catheter infections were statistically significantly greater in patients who were newly diagnosed during that year, compared to previously identified cases whose care was continuing [$2713.4 (2486.2–2940.5) in new cases compared to $5481.7 (5263.8–5699.6) in old cases *p* = 0.000 for HD treatment; $450.5 (425.1–475.8) in new cases compared to $671.3 (648.03–694.5) in old cases, *p*= 0.000 for control costs of HD treatment; and $56.60 (21.76–91.44) in new cases compared to $427.8 (222.5–633.1) in old cases, *p* = 0.007 for the inpatient treatment costs of catheter infections]. The cases newly diagnosed during that year were compared to cases diagnosed before the specific study period. A total of 70 patients were newly diagnosed with ESRD in 2019 out of a total of 310 patients, with 73 new cases identified in 2020 out of a total of 326 patients (22.6% of cases newly diagnosed in 2019 compared to 22.4% of cases newly diagnosed in 2020). The HD treatment costs, HD control costs, and inpatient treatment costs of catheter infections were statistically significantly greater in patients newly diagnosed in 2020 compared with those patients whose care for ESRD was continuing $2089.0 [(1906.0–2272.0) in new cases compared to $4287.8 (4154.8–4420.7) in old cases, *p* ≤ 0.001 for the cost of HD treatment; $317.1 (298.06–336.1) in new cases compared to $523.6 (510.12–537.08) in old cases, *p ≤* 0.001 for the control costs of HD patients; and $64.52 (37.94–91.10) in new cases compared to $152.4 (87.92–216.88) in old cases for the inpatient treatment costs of catheter infections, (*p* = 0.033)].

## 4. Discussion

### 4.1. Main Findings of the Study

COVID-19 cases were only detected in patients having HD treatment. A statistically significant difference was found between the PCR and CT mean costs in patients receiving RRT, with the highest in the HD patients. When the costs of 2019 and 2020 were compared, a statistical difference was found in the costs of HD treatment and laboratory costs.

### 4.2. Explanation of Study Findings and Comparison with Existing Literature

The present study showed that among the 358 ESRD patients receiving RRT, 13 patients had COVID-19 infections, with the cases only observed in haemodialysis patients, and there were no cases detected in PD and Tx patient groups. Previous research reported HD patients might have increased risk of having COVID-19 infection, compared to the general population [17]. The mortality rate in HD patients was high and 38.5% of HD patients with COVID-19 died while having ICU treatment in the hospital. This mortality rate was 14.4 times greater than the mortality rate in the general population (2.7%). Previous research also showed high rates of death in HD patients, with mortality rates of COVID-19 ranging from 15% to 41% [15,18,19,20,21,22]. The high rates of mortality in the HD group could be because this group included those with a higher average age compared to the general population [23,24] The higher mortality rates in the HD population could be explained by the multiple comorbidities present in the group, such as diabetes, hypertension, and other cardiovascular conditions [25]. The majority of the ESRD population were vaccinated, with over 90% of patients having third or fourth doses of COVID-19 vaccines. The lockdown and other COVID-19 prevention policies were applied very strictly in the first year of the pandemic. These might explain the reduced risk of COVID-19 infection in the PD and Tx patients who had not attended hospital as frequently as the HD patients did.

The current study found that the cost of care given to HD patients was the highest among the different types of RRT. This finding is consistent with the findings of previous research, which showed that HD costs were higher than the costs of Tx and PD [26,27,28]. HD in Northern Cyprus is only provided in hospitals, with no application of home-based HD in the area. The existing research outlines that HD provided in hospitals exerts higher costs of care, compared to home-based HD, even in the initial year of the RRT [29]. Previous studies showed that HD treatment costs were higher than health expenditure spent per capita, which revealed the need for the provision of effective and cost-effective strategies for preventing ESRD and alternative cost-effective methods of RRT, such as the enhancement of Tx [30,31], encouragement of PD in eligible patients and home-based dialysis [29,32].

The present study showed that the risk of COVID-19 infection transmission and mortality was highest in patients receiving HD. This could be explained by the fact that the need to commute to the hospital and have treatment for long hours in haemodialysis centres led to an increased risk of infection. The increased mortality in HD patients could be due to comorbidities and the older age of the specific patient group. Therefore, this makes it essential to take necessary precautions, such as the development of effective isolation mechanisms for patients with COVID-19 [33], making arrangements for the use of HD machines only by COVID-19 patients, enhancing the use of PD in eligible patients, and finding alternative methods for providing home-based dialysis care, particularly for patients with comorbidities [32].

Catheter infections in HD patients are determinants of morbidity and mortality in HD patients [34]. It is important to analyse the rates of catheter-related infections in HD patients. Unlike previous research, our study showed that the rates of catheter infections were higher after the COVID-19 pandemic in 2020, compared to 2019 [35]. Heidempergher [35] and Johansen [36] suggested that the rates of catheter infections reduced during the pandemic due to the standard hygiene procedures applied. However, in Northern Cyprus, the catheter infection rates increased, which might suggest insufficient or inappropriate hygiene procedures, and this should be followed up for the prevention of transmission of COVID-19 in HD patients. The outpatient treatment costs due to catheter infections were not significantly different before and after the COVID-19 pandemic; however, there was an increase in the inpatient treatment costs of catheter infections. This could be explained by the fact that, during the pandemic period, there were attempts to minimise hospitalisations to prevent the risk of the COVID-19 transmissions in HD patients.

### 4.3. Strengths and Limitations of this Study

This study provides important implications for the management of ESRD patients, particularly during a pandemic, an important type of crisis that can significantly impact the management of patients with chronic conditions. This study firstly outlined the practices that were implemented to effectively manage ESRD. The allocation of dialysis machines to COVID-19 patients and those in isolation were important practices designed for the care of ESRD patients. The study further determined the cost burden of patients undergoing RRT. The type of dialysis, habits, and comorbidities were important determinants of the direct cost of ESRD treatment, and analysing them was important in terms of allocating resources based on demand [7].

The study showed the impact of infections on the increasing costs of RRT, since infections were among the major risk factors. It is important to study the cost of RRT due to infections to ensure policymakers efficiently allocate resources for infection management and prevention. There is a lack of studies reporting the cost burden of RRT in kidney patients, so this was the first study to research the cost burden of RRT in Northern Cyprus. One of the limitations of the study was that the indirect costs could not be studied, because data regarding such costs were difficult to retrieve. At the beginning of the pandemic, there were very strict precautions taken in the country because of the limited resources available, such as absence of a pandemic hospital, limited healthcare staff, limited equipment, among others. Since the number of cases were very low, even in the whole population, in the first six months after the first outbreak, an analysis with sub-periods would not allow accurate variability analysis between subgroups. Multicentre analysis, to show variability in costs between centres, was not conducted, because the patients registered in centres other than centre A received the majority of their dialysis and complicated treatment processes in centre A in the pandemic period.

## 5. Conclusions

Efforts to study the control measures for improving ESRD care in populations are crucial, since it is a condition that leads to morbidity and a high-cost burden. This study presented important implications for the care of ESRD patients, particularly during the outbreak of infections, such as COVID-19. HD is the most frequently applied RRT in Northern Cyprus, which leads to a high risk of morbidity and cost-burden, compared to other RRT. A risk of COVID-19 was shown to be associated with HD, along with the high costs of diagnosis and management costs required for COVID-19. It was further shown that the risk of death, due to COVID-19, was also higher in HD patients compared to patients receiving other RRT. This reveals the need for cost-effective and efficient methods of managing ESRD, such as encouraging PD in eligible patients, enhancing Tx by developing targeted policies or encouraging the use of home-based dialysis.

## Figures and Tables

**Table 1 healthcare-10-02351-t001:** Characteristic information of End-Stage Renal Disease patients receiving Renal Replacement Therapy.

Variables	Groups	Year			
		2019		2020	
		N Number	% Frekans	n Number	% Frekans
ESRD Patients	Haemodialysis	252	81.3	257	78.83
	Peritoneal dialysis	29	9.4	30	9.20
	Transplantation	29	9.4	38	11.65
	Total	310	100	326	100
According to RRT patient number	Only HD	252	81.3	257	78.83
	Total	310	100	326	100
Gender	Female	111	35.8	112	34.4
	Male	199	64.2	214	65.6
	Total	310	100	326	100
Situation	Live	253	81.6	277	85.0
	Ex	57	18.4	49	15.0
	Total	310	100	326	100
The cases identified in the particular year	2019	70	22.6	-	-
	2020	-	-	73	22.4
Dialysis machines numbers of centres	A centre	33	49.25	33	49.25
	B centre	7	10.44	7	10.44
	C centre	11	16.41	11	16.41
	D centre	10	14.92	10	14.92
	Reserve	6	8.95	6	8.95
	Total	67	100	67	100
HD Patient number only main Centre	HD patient	146	47.09	169	51.84
	Total	310	100	326	100
Patient with catheter infections in main centre	infected patient for HD patient	34	23.28	67	39,64
	Total	146	100	169	100

**Table 2 healthcare-10-02351-t002:** Dialysis machines capacity of dialysis centres in 2020.

Centers	Groups	N (Number)
A Centre	Surgical intensive care	1
	Triage intensive care	1
	HCV RNA (+)	1
	HIV	1
	HBS	1
	HCV RNA (−)	1
	Reserve	6
	İsolated machine for contact patients	1
	For emergency	1
	Covid (+) patients	1
	For normal regular patients	23
	Total	39
B Centre	Normal regular patients	7
	Total	7
C Centre	For normal regular patients	8
	HCV RNA (+)	1
	HBS	1
	Reserve	1
	Total	11
D Centre	For normal regular patients	10
	Total	10

**Table 3 healthcare-10-02351-t003:** Number of COVID-19 cases detected in patients receiving haemodialysis between March 2020, and October 2021.

Groups of RRT Methods	N (Number)	COVID-19 Case Number
Haemodialysis	285	13
Peritoneal Dialysis	29	-
Transplantation	44	-
Total	358	13

**Table 4 healthcare-10-02351-t004:** The relationship of renal replacement therapy methods with polymerase chain reaction and computer tomography costs between March 2020–March 2021($).

Variable	Groups	N	PCR Cost ($)	CT Cost ($)		Variable	Groups	
			Mean (95% CI)		CV	Mean (95% CI)		CV
**RRT methods**	HD patients	257		75.53 (67.72–83.34)	0.84		3.51 (2.71–4.31)	0.84
	PD patients	29		36.75 (23.8–50.12)	0.95		1.00 (−0.23–2.23)	0.95
	Tx patients	40		8.67 (3.82–13.52)	1.74		0.17 (−0.17–0.52)	1.74
	Total patients	326		63.88 (57.10–70.65)	0.97		2.88 (2.22–3.53)	0.97
				F: 26,680			F: 27,380	
				*p* < 0.001			*p* < 0.001	

PD: Peritoneal dialysis; Tx: Transplantation; PCR: polymerase chain reaction; CT: Computer tomography; $: Dollars; CV: the variation of coefficient.

**Table 5 healthcare-10-02351-t005:** Direct medical cost of studied patients between 2019–2020 and between 2020–2021.

Variables of Costs	Year				
	2019 Total HD Patient = 252 * Infected Patient = 34		2020 Total HD Patient = 257 * Infected Patient = 63		*p*-Value
	Mean (95% CI)	CV	Mean (95% CI)	CV	
** HD treatment cost	4822.65 (4557.89–5987.43)	0.44	3759.45 (3.573–3945.8)	0.40	≤0.001
*** Control cost of HD patients	618.80 (592.2–645.3)	0.34	474.03 (455.53–492.54)	0.31	≤0.001
Outpatient treatment cost of catheter infection	54.61 (34.29–74,94)	1.02	54.74 (40.72–68.76)	1.01	>0.793
Inpatient treatment cost of catheter infection	315.33 (131.34–499.32)	1.64	121.03 (66.28–175.78)	1.78	≤0.015

* Infected patients, whose treatment costs are calculated, consist only of haemodialysis patients followed in the main hospital. ** Each bicarbonate fee was taken in calculating the cost of HD treatment. HD sessions are applied to the patients on average 3 times a week. *** In general laboratory cost calculations, routine imaging and laboratory tests were calculated for each HD patient. The total cost of catheter infection includes the cost of outpatients and inpatients. In addition, the cost of patients with catheter infection includes laboratory tests, imaging, medication and hospital-bed care fees. CV: the variation of coefficient.

**Table 6 healthcare-10-02351-t006:** Comparative analysis of characteristics information and costs values ($) of two different time periods (in 2019 and in 2020).

Characteristic	Year	Groups	* HD Treatment Cost		* Control Cost of HD Patients		** Outpatient Treatment Cost of Catheter Infections		** Inpatient Treatment Cost of Catheter Infections	
			Mean (95% CI)	CV	Mean (95% CI)	CV	Mean (95% CI)	CV	Mean (95% CI)	CV
The time of the diagnosis of ESRD	2019	Cases diagnosed before 2019	5481.7 (5263.8–5699.6)	0.32	671.3 (648.03–694.5)	0.28	60.4 (37.78–83.02)	1.06	427.8 (222.5–633.1)	1.37
		Newly diagnosed cases in 2019	2713.4 (2486.2–2940.5)	0.67	450.5 (425.1–475.8)	0.45	42.45 (27.81–57.09)	0.98	56.60 (21.76–91.44)	1.75
		*p*-value	0.000		0.000		0.408		0.007	
	2020	Cases diagnosed before 2020	4287.8 (4154.8–4420.7)	0.25	523.6 (510.12–537.08)	0.21	57.6 (42.8–72.40)	1.01	152.4 (87.92–216.88)	1.67
		Newly diagnosed cases in 2020	2089.0 (1906.0–2272.0)	0.71	317.1 (298.06–336.1)	0.49	48.33 (35.0–61.66)	1.09	64.52 (37.94–91.10)	1.63
		*p*-value	≤0.001		≤0.001		0.279		0.033	

* HD patient number (in 2019 n = 252, in 2020 n = 257); ** patient with catheter infection (in 2019 n = 34, in 2020 n = 63); CV: the variation of coefficient.

## Data Availability

The data that support the findings of this study are available from the corresponding author upon reasonable request.

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
