# Peer review of "Risk of COVID-19 and Cost Burden in End-Stage Renal Disease Patients and Policy Implications for Managing Nephrology Services during the COVID-19 Pandemic"

_healthcare, 2022, doi:10.3390/healthcare10122351_

Round 1

Reviewer 1 Report

COMMENTS TO THE AUTHORS of Manuscript ID: healthcare-2023905

The paper presented under the title "Risk of COVID-19 and cost burden in end-stage renal disease patients and policy implications for managing nephrology services during the COVID-19 pandemic”

I indicate in red the clippings of the original text.

1.-The authors define the objective and the hypotheses of the research in points “.1 and 2.2

2.1. Study Design and Data Collection

The study was carried out as a retrospective cohort study, which included data collected from the patient’s records of 4 different dialysis centres providing ESRD care for patients. The data for the study were collected from the clinical, administrative, and financial records of hospitals affiliated to the Health Ministry in Northern Cyprus.

2.2. The Research Questions and Significance of the Study

The research questions and motivation factors were as follows:

• What are the costs of COVID-19 diagnostic methods such as PCR and CT?

• What are tue costs of COVID-19 diagnostic methods according to RRT?

• What was the incidence of COVID-19 in ESRD patients between March 2020- 119 October 2021?

• What were the HD treatment costs, laboratory costs, and catheter infections costs of HD patients in 2019 and 2020, comparatively before and after the outbreak of the COVID-19 pandemic?

-In my opinion, I think they should give more information about the cohorts. Because at the beginning of the pandemic it had nothing to do with the end of the pandemic, and this has not been taken into account. The authors could have made a partition of time and observed what happens in 4 moments, in this way, it would be possible to see how the measures they analyze evolved. I'm not saying that they do, I'm just saying that they could mention it, in the discussion at least. (See Livacic-Rojas, et al., 2010; Vallejo et al., 2019).

2.-I think that tables 4, 5, and 6 should be presented in another way. That is, there is too much information, and the results look very bad. In my opinion, I would leave only the result in dollars in the text, and in complementary material, I would put the result in another currency, so that interested people go to see the complementary material. The expression in dollars is better understood by everyone since it is a reference currency. That is why I think that in the text it should only be in dollars. This way the result of the tables will look lighter, and the information will be captured better.

3.-The authors should explain the reason for the great variability in the data. The SD is greater than the mean on many occasions, which means that the distribution of the variable is not normal. In addition, there is a great difference between the variances of the three groups, HD, PD, and Tx, and in the analysis of the data, the authors do not say anything about this. The authors should report whether or not the assumptions of normality and homogeneity are met, and should proceed accordingly, transforming the data, using robust parametric tests, or using non-parametric statistics. Authors should reanalyze the data in absolute mode. SPSS offers several possibilities. The way the authors analyze the data is not correct.

4.-The authors must review the data presented in Table 5, because I, at least, have noticed some errors in the figures.

5.-The authors should also expose the variation coefficient, it adds information and allows comparing the result of different variables or in different samples.

6.-Finally, I suggest that in the conclusions section, the authors indicate that a desirable way to carry out this study is to be able to have multiple centers to be able to carry out a multilevel analysis and to be able to study to what extent the variability between the centers determines the differences in the variables they are analyzing. (see Vallejo et al., 2013).

References:

Livacic-Rojas, P., Vallejo, G., & Fernández, P. (2010). Análisis de las tasas de error de tipo I de procedimientos univariados y multivariados en diseños de medidas repetidasCommunications in Statistics—Simulation and Computation®39(3), 624-640.

Vallejo, G., Ato, M., Fernández, M. P., & Livacic-Rojas, P. E. (2019). Estimación del tamaño de la muestra para modelos heterogéneos de curvas de crecimiento con desgaste. Behavior Research Methods51(3), 1216-1243. https://doi.org/10.3758/s13428-018-1059-y

Vallejo, G., Ato, M., Fernández, P., & Livácic, P. (2013). Análisis bootstrap multinivel con supuestos incumplidos. Psicothema. 25(4):520-8. dos: 10.7334/psicothema2013.58.

Author Response

We are grateful for your valuable time for reviewing our manuscript. The point-by-point responses can be found in the attached document. 

Reviewer 2 Report

Overall, this manuscript analyzes the infection risk of COVID-19 infection in ESRD patients and the cost burden of treatment and management of ESRD during COVID-19 pandemic period. Although the data put forward some concepts, there remains several unanswered questions. Most importantly is that the results do not strongly support their conclusions. Major concerns are listed below:

1.     The major conclusion of this manuscript is that ESRD patients on HD have a higher incidence rate compared to ESRD patients receiving PD or Tx, and the general population. This conclusion is very doubtful due to the small sampling numbers of patients.  This conclusion is only based on data from table 3. First, wrong calculation, 13 COVID-19 case out of 285 patients make the incidence rate 4.56% not 5.03%. Second, it is not surprised that no COVID-19 case reported in ESRD patients on PD or Tx, with such small sampling numbers of 29 and 44 for PD and Tx patients, respectively. This is statistically doubtful. In one hand, the current results show incidence rate of COVID-19 is 0 for PD or Tx patients. In reality, this can not be true, otherwise we will get a conclusion that ESRD patients with PD and Tx have way lower risk for COVID-19 compared to the general population, and PD and Tx can protect ESRD patients from COVID-19 infection. On the other hand, image that if 2 COVID-19 cases reported in 29 PD patients, which gives the highest incidence rate of 6.89% and then we get a totally different conclusion.

This manuscript collects plenty data for the comparison of cost for treatment and management of ESRD patients before and after COVID-19 outbreak. It is surprised that the overall cost of cares given to ESRD patients is lower after COVID-19 breakout compared to before COVID-19. However, the authors do not explain the reason or discuss any implications or guide which this lower cost can give to current or future treatment and management for ESDR patients.  Another conclusion of this manuscript is that HD costs were higher than the costs of Tx and PD. However, the higher cost for HD is not related to COVID-19 outbreak. Whenever before or after COVID-19, HD always has a higher cost compared to PD and Tx. The conclusion that A risk of COVID-19 was shown to be associated with HD, is not convincing.

Author Response

(The authors gave the same response as above.)

Reviewer 3 Report

This is a descriptive study that discusses how the COVID pandemic has changed the management of patients with ESRD with special attention to the financial consequences.

Minor comments

-          How many patients were vaccinated? Can we have that data as well?

-          How many patients died due to COVID-19 versus non-COVID reasons

-          Please consider these articles for discussion:

https://pubmed.ncbi.nlm.nih.gov/33465417/

https://pubmed.ncbi.nlm.nih.gov/36188740/

Author Response

(The authors gave the same response as above.)

Reviewer 4 Report

The manuscript “Risk of COVID-19 and cost burden in end-stage renal disease patients and policy implications for managing nephrology services during the COVID-19 pandemic” presented very systematically the evaluation of the risk of COVID-19 in ESRD patients in Northern Cyprus (data collected from 4 dialysis centers), the cost for monitoring COVID-19 transmission in patients having ESRD, the cost of haemodialysis treatment, laboratory costs and catheter infections costs of haemodialysis patients in 2019 and 2020, before and after the COVID-19 pandemic outbreak. It is a new study involving data processing from the records of 4 dialysis centers in Northern Cyprus. This study provides important implications for the management of ESRD patients during pandemic crisis and the cost burden of patients undergoing of renal replacement therapy including haemodialysis which is the most frequently applied in Northern Cyprus. The main conclusions of the study contribute to the subject area previously published by researcher from USA, Brazil, Spain, England, France, Netherland, Italy, Turkey, Qatar, Ethiopia, and China. The conclusions are consistent with the evidence and arguments presented. They addressed all the main research questions of the study. The references cited in the article are recently published and closely related to the topic. The manuscript is very well written. The use of English language is very good, and the sections are easy to follow.

Minor technical error: in line 177 add 34. for the last reference.

Author Response

(The authors gave the same response as above.)

Round 2

Reviewer 2 Report

The authors addressed my concerns and I have no more comments on the manuscript.